# Large Cancer Pedigree Involving Multiple Cancer Genes including Likely Digenic *MSH2* and *MSH6* Lynch Syndrome (LS) and an Instance of Recombinational Rescue from LS

**DOI:** 10.3390/cancers15010228

**Published:** 2022-12-30

**Authors:** Ingrid P. Vogelaar, Stephanie Greer, Fan Wang, GiWon Shin, Billy Lau, Yajing Hu, Sigurdis Haraldsdottir, Rocio Alvarez, Dennis Hazelett, Peter Nguyen, Francesca P. Aguirre, Maha Guindi, Andrew Hendifar, Jessica Balcom, Anna Leininger, Beth Fairbank, Hanlee Ji, Megan P. Hitchins

**Affiliations:** 1Department of Medicine (Oncology), Stanford Cancer Institute, Stanford University, Stanford, CA 94305, USA; 2School of Public Health (Epidemiology), Harbin Medical University, Harbin 150088, China; 3Bioinformatics and Functional Genomics Center, Department of Biomedical Sciences, Cedars-Sinai Medical Center, Los Angeles, CA 90048, USA; 4Department of Pathology and Laboratory Medicine, Cedars-Sinai Medical Center, Los Angeles, CA 90048, USA; 5Samuel Oschin Cancer Center, Cedars-Sinai Medical Center, 8700 Beverly Blvd, Los Angeles, CA 90048, USA; 6Department of Laboratory Medicine and Pathology, Mayo Clinic, Rochester, MN 55902, USA; 7Minnesota Oncology, Woodbury, MN 55125, USA; 8Lynch Syndrome Australia, The Summit, QLD 4377, Australia; 9Stanford Genome Technology Center West, 1050 Arastradero, Palo Alto, CA 94304, USA; 10Lowy Cancer Research Centre, University of New South Wales, Sydney, NSW 2052, Australia

**Keywords:** Lynch syndrome, splice variants, multicancer gene panel test

## Abstract

**Simple Summary:**

Lynch syndrome is a hereditary cancer condition caused by a pathogenic variant (mutation) within one of the mismatch repair (MMR) genes. Risks for cancer vary by which MMR gene is mutated and sex, although colorectal and uterine cancers are most common. A major challenge in genetic testing is that this frequently reveals a variant of uncertain significance (VUS), which hinders clinical decision-making. We describe a large, four-generation, 13-branched, cancer-affected family with a mutation in *MSH2* (c.2006G>T) in which one branch also carries a VUS in *MSH6* (c.3936_4001+8dup). Functional studies in samples from family members show this *MSH6* VUS is likely to be pathogenic. In addition, other cancer-relevant mutations were identified in branches without either MMR gene mutation so genetic counseling was highly individualized. Our study suggests multi-cancer gene panel testing should be offered to all members of cancer-effected families rather than targeted testing for specific mutations for accurate genetic diagnosis.

**Abstract:**

Lynch syndrome (LS), caused by heterozygous pathogenic variants affecting one of the mismatch repair (MMR) genes (*MSH2*, *MLH1*, *MSH6*, *PMS2*), confers moderate to high risks for colorectal, endometrial, and other cancers. We describe a four-generation, 13-branched pedigree in which multiple LS branches carry the *MSH2* pathogenic variant c.2006G>T (p.Gly669Val), one branch has this and an additional novel *MSH6* variant c.3936_4001+8dup (intronic), and other non-LS branches carry variants within other cancer-relevant genes (*NBN*, *MC1R*, *PTPRJ*). Both *MSH2* c.2006G>T and *MSH6* c.3936_4001+8dup caused aberrant RNA splicing in carriers, including out-of-frame exon-skipping, providing functional evidence of their pathogenicity. *MSH2* and *MSH6* are co-located on Chr2p21, but the two variants segregated independently (mapped *in trans*) within the digenic branch, with carriers of either or both variants. Thus, *MSH2* c.2006G>T and *MSH6* c.3936_4001+8dup independently confer LS with differing cancer risks among family members in the same branch. Carriers of both variants have near 100% risk of transmitting either one to offspring. Nevertheless, a female carrier of both variants did not transmit either to one son, due to a germline recombination within the intervening region. Genetic diagnosis, risk stratification, and counseling for cancer and inheritance were highly individualized in this family. The finding of multiple cancer-associated variants in this pedigree illustrates a need to consider offering multicancer gene panel testing, as opposed to targeted cascade testing, as additional cancer variants may be uncovered in relatives.

## 1. Introduction

Lynch syndrome (LS) is the most common hereditary colorectal cancer sysndrome, caused by a heterozygous germline pathogenic variant (PV) within one of the DNA mismatch repair (MMR) genes *MLH1* (MIM 609310)*, MSH2* (MIM 120435), *MSH6* (MIM 614350), *PMS2* (MIM 600259) [1] or deletion of the 3′ end of the *EPCAM* gene leading to epigenetic inactivation of neighboring *MSH2* (MIM 185535) [2,3]. LS is autosomal dominant with 50% risk of inheritance of the cancer-predisposing PV. Typically, a second somatic event affecting the wild-type allele leads to the development of a mismatch repair deficient (dMMR) cancer exhibiting microsatellite instability (MSI), most frequently colorectal cancer (CRC) and endometrial cancer (EC) [1]. Constitutional mismatch repair deficiency (CMMRD), caused by biallelic (homozygous or compound heterozygous) pathogenic variants in one of four MMR genes, is rare and more severe, characterized by very early (typically pediatric) onset of CRC, hematological malignancies, glioblastoma, and urinary tract tumors [1]. CMMRD has been described in consanguineous families in which both parents have LS due to carriage of the same heterozygous germline mutation [4]. Their offspring thus have a 25% risk of CMMRD and 50% risk of LS.

Persons with LS are at high life-risk of developing cancer, ranging from 34–84% depending on gene affected and sex [5,6]. Although CRC and EC are the most frequent syndromic cancers, LS can involve a wide spectrum of cancers including ovarian, other gastrointestinal, skin, renal and urinary tract, brain, prostate, and breast cancers [1]. Carriers of a PV within *MLH1* or *MSH2* have an increased risk of the full spectrum of LS-associated cancers with especially high life-time risks for CRC (up to 57%) and EC (up to 50%) [5,7]. Carriers of a PV in *MSH6* and *PMS2* have a substantially lower risk of CRC and the age at diagnosis is over 10 years later than for carriers of a PV in *MLH1* or *MSH2* [8]. The risk of developing other extracolonic cancers is also much lower in carriers of a PV in *MSH6* or *PMS2* compared to *MLH1* and *MSH2* [5,9,10,11], with one major exception: female carriers of a PV in *MSH6* have high life-time risks for gynecological cancers (41% for EC, up to 11% for ovarian cancer), comparable to carriers of a PV in *MLH1* or *MSH2* [8]. To address these differing cancer risks, a new classification system for LS has been proposed based on genetic diagnosis, in which the underlying gene affected is designated (e.g., *MSH2*-LS), and clinical surveillance recommendations are tailored accordingly [5]. Persons with *MLH1*-LS or *MSH2*-LS are advised to undergo colonoscopy every 1–2 years from age 20–25 years and have additional surveillance of the endometrium, ovaries, and urinary tract, whereas persons with *MSH6*-LS or *PMS2*-LS are advised to have a colonoscopy every 1–2 years from age 25–30 years and additional surveillance of the endometrium in females [5].

An ongoing challenge for genetic diagnosis of LS is the identification of “variants of uncertain significance” (VUS) within the MMR genes in probands with a clinical suspicion. Classification of the pathogenicity status of novel genetic variants involves various integrated factors, including clinical diagnoses, pathology and molecular profile of tumors, population frequency, and functional effect on gene products.

Herein, we describe a large cancer-affected pedigree bearing the germline PV *MSH2* c.2006G>T (p.Gly699Val) in several branches, in which one branch additionally carries a novel germline *MSH6* variant c.3936_4001+8dup (Intronic). At the time these variants were identified in this family, both were classified as VUS. We evaluated the contribution to cancer risk by each of the two MMR variants in this family by studying the molecular characteristics of available colorectal adenocarcinoma and adenomas from carriers, segregation analyses with diagnoses of the major LS cancers in the pedigree, cumulative risk for the onset of cancer and preneoplastic colorectal polyps, and their functional impact on mRNA transcripts. Functional mRNA analyses on biospecimens from carriers of either or both variants indicate that both *MSH2* c.2006G>T and *MSH6* c.3936_4001+8dup are pathogenic splice variants predicted to result in loss of protein function, which for *MSH2* c.2006G>T is consistent with previously reported findings [12,13]. Extended genetic screening revealed additional germline variants within the cancer-associated *NBN*, *MC1R*, and *PTPRJ* genes in non-LS branches of the family in which several members also had cancer diagnoses. Our study suggests cascade genetic testing involving comprehensive (cancer gene panel) testing should be considered to provide a full and accurate genetic diagnosis, individualized post-test counseling, cancer and inheritance risk stratification, and clinical surveillance recommendations, given the risks for genetic inheritance and cancer may differ within families with more than one cancer-associated genetic variant.

## 2. Subjects and Methods

### 2.1. Recruitment of Family Members

Family members were approached by the proposita and provided with the contact details of the study investigators with the option of self-referral to join the study. All subjects provided their written, informed consent to join the study and use their health information for research purposes. This study was approved by the Human Research Ethics Committee of the University of New South Wales, Sydney, Australia, and by the Institutional Review Boards of Cedars-Sinai Medical Center (#Pro00049624), Los Angeles, and Stanford University (#40382), Stanford, California, both in California, USA. All consenting subjects were interviewed in person or by phone and were asked about their personal and cancer family histories. These were corroborated across family members and, where possible, clinical details were verified from medical records, pathology reports, prior genetic testing reports, and communications with their treating physicians, upon permission by each subject. For family members interested in undergoing formal genetic testing at a CLIA-approved laboratory, pre- and post-test genetic counseling was arranged.

### 2.2. Specimen Collection and Processing

Living participants provided a sample of saliva and/or peripheral blood and/or hair follicles for in-house DNA and RNA analyses. Samples collected are listed in Appendix A. Saliva and hair follicles were self-collected remotely. Saliva samples were collected in Oragene OGR-500 DNA and Oragene RE-100 RNA self-collection tubes (DNA Genotek, Ottawa, ON, Canada). From saliva, DNA was extracted using the prep IT.L2P kit (DNA Genotek, Ottawa, ON, Canada), and RNA was extracted using the RNeasy Micro Kit (Qiagen, Valencia, CA, USA). Peripheral blood samples were collected in K_2_EDTA and/or DNA/RNA Shield (Zymo Research, Irvine, CA, USA) and/or acid citrate dextrose (ACD) solution tubes at the participant’s clinical treatment center. All samples were shipped at ambient temperature on overnight delivery to the respective laboratory. Peripheral blood mononuclear cells (PBMCs) were isolated from fresh blood samples collected in K_2_EDTA or ACD tubes using Ficoll reagent (Ficoll-Paque; GE Healthcare, Piscataway, NJ, USA) according to the manufacturer’s protocol. DNA and RNA were extracted from whole blood, PBMCs, and hair follicles using the Blood and Cell Culture DNA Midi Kit and the RNeasy Mini Kit (Qiagen, Valencia, CA, USA), respectively. Nucleic acid purity and quantification were determined by spectrophotometry. For participants undergoing CLIA-approved diagnostic genetic testing, a separate saliva sample was self-collected and sent directly to Invitae for testing by next-generation sequencing (NGS) on the 134 cancer-gene panel.

### 2.3. Pathology and Immunohistochemistry of Formalin-Fixed Paraffin-Embedded (FFPE) Tissues

FFPE tissue blocks and accompanying pathology reports were obtained from treating hospitals following the consent and sign of a release-of-information form by the family member or next-of-kin. Diagnosis, AJCC stage, MSI, and immunohistochemical expression status of the MMR proteins (where previously conducted) was obtained from pathology reports. A 4 µM section of each FFPE tissue block was stained with hematoxylin and eosin (H&E). Immunohistochemistry (IHC) of all four MMR proteins was performed on serial 4 µM sections either as described below or previously [14]. All IHC staining protocols were fully automated with antigen retrieval. Mouse monoclonal antibodies included α-MLH1 and α-PMS2 (G168-15 and A16-4, respectively, from BD Pharmingen, San Diego, CA), α-MSH2 (G219-1129, Cell Marque, Rocklin, CA, USA), and α-MSH6 (clone 44, Ventana, Tucson, AZ, USA). IHC detection was performed on the Ventana benchmark Ultra instrument with Ultraview DAB detection for MLH1, MSH2 and MSH6 and the Leica Bond III instrument with Leica refine DAB detection for PMS2. All slides were subsequently counterstained with Mayer’s hematoxylin. IHC interpretation, histology, and demarcation of areas of highest tumor cellularity representative of adenocarcinoma and areas of NCM on each H&E slide was performed by a gastrointestinal pathologist. With reference to the H&E slide, a 1 mm diameter core of each tissue type was excised from the respective demarcated areas on each block. Each FFPE core was deparaffinized in 1 mL Deparaffinization Solution (Qiagen), then DNA extracted using the QIAamp FFPE DNA Kit (Qiagen) according to the manufacturer’s instructions.

### 2.4. In-House Testing of the MMR Genes for Genetic Mutations and Methylation

To screen for single nucleotide variants and small insertion-deletions, the exons and intron-exon boundaries of *MLH1*, *MSH2*, and *MSH6* genomic DNA from peripheral blood leukocytes (PBL) was amplified using intronic primers and cycling conditions as previously described [15]. PCR products were purified with the QIAquick PCR purification kit (Qiagen, Valencia, CA, USA) and bidirectionally Sanger sequenced on the ABI 3730XL DNA Analyzer (Applied Biosystems, Foster City, CA, USA) with BigDye terminator methodology using the original PCR primers. Sequences were analyzed using Sequence Scanner software 2. Multiplex ligation probe analysis (MLPA) was performed to detect larger deletions or duplications using the SALSA MLPA Probemixes (MRC Holland, Amsterdam, The Netherlands) P003 for *MLH1*, *MSH2*, and *EPCAM*, and P072 for *MSH6*, *EPCAM*, and the common European *MUTYH* variants c.536A>G and c.1187G>A, both according to the manufacturers’ standard operating procedures. Fragment analysis was conducted on the 3730XL DNA Analyzer using GeneMapper software (Applied Biosystems, Waltham, MA, USA). Dosage analyses were performed using Coffalyser software alongside genomic DNA from healthy controls.

For targeted testing of the *MSH2* variant c.2006G>T (p.Gly669Val) and the *MSH6* variant c.3936_4001+8dup (Intronic) in relatives, sequences encompassing the genetic variants were PCR-amplified (Appendix A), purified, and sequenced as above. To initially resolve the *MSH6* c.3936_4001+8dup (Intronic) variant, the PCR products from three carriers were cloned using the TOPO^®^ TA Cloning^®^ Kit for Sequencing (ThermoFisher Scientific, Grand Island, NY, USA) and Sanger sequenced using Sanger M13 vector primers. GenBank accession numbers for reference sequences were NM_000251.2 for *MSH2* and NM_000179.2 for *MSH6*, consistent with the Leiden Open Variation Database v.3.0. Build 25b curated by InSiGHT (available at https://www.insight-group.org/variants/databases/ accessed on 3 March 2017).

*MSH2* promoter methylation levels were measured using a CpG pyrosequencing assay as previously described [14]. The CpGenomeTM Human Methylated and Non-methylated DNA Standards (Millipore, Burlington, MA, USA) were used as hypermethylated and unmethylated controls, respectively. Samples with average methylation levels > 2.5% (background noise) were considered methylation-positive.

### 2.5. Co-Segregation Analyses of MSH2 and MSH6 Variants with Disease Status

To measure the frequency of a germline variant and disease inherited together, we utilized COOL (Co-segregation Online, http://fengbj-laboratory.org/cool2/, accessed on 19 January 2021) to perform a Bayes factor (BF) co-segregation analysis [16]. Using R, the PedigreeXP output file used to generate the pedigree (Figure 1) was converted to a file format that contains the following features necessary for the analysis: (1) Individual ID (2) Father ID (3) Mother ID (4) Sex (5) Affection status (6) Age (7) Proband and (8) Genotype. Affected status and age of cancer-affected were based on the earliest age of onset up to the end of the observation period (30 April 2021). Finally, pseudo-parents were added only when one parent was known. The affection status and genotype of these individuals were set to unknown. The age was inferred from the oldest child age plus 25.

### 2.6. Risk for Cancer Onset by Carrier Status for Variants MSH2 c.2006G>T, MSH6 c.3936_4001+8dup (Intronic), or Both

Self-reported colorectal polyp and cancer (any type) data were analyzed using the IBM Statistical Package for the Social Sciences (SPSS) version 21 (IBM Corporation, Armonk, NY, USA) and R statistical package. The observation time was from birth until the year of first diagnosis of any cancer or of pre/neoplasm (cancer and polyp(s) pooled) or the end of the observation (30 April 2021). Kaplan–Meier was used to generate the cumulative risk curve and life tables were used to estimate the cumulative proportion and its standard error of first cancer or pre/neoplasm occurrence at different ages. Statistical significance between any group was sought using the log-rank test and considered significant when *p* ≤ 0.05 [17].

### 2.7. RNA Analyses to Detect and Measure Aberrant Splicing

To examine effects of the variants on mRNA splicing, complementary DNA (cDNA) was synthesized from total RNA by reverse-transcription (RT) using SuperScript^®^ III First-Strand Synthesis SuperMix (ThermoFisher Scientific) and PCR amplification of cDNA was performed across exons 12–14 for *MSH2* and exons 8–10 for *MSH6*, followed by gel electrophoresis for sizing (Appendix A). The relative intensities of altered (shorter or longer length) fragments versus full-length transcripts on the electrophoresis gel was measured by densitometry using the Image Lab 5.0 (Bio-Rad laboratories, Hercules, CA, USA). Relative fragment intensity was calculated as a ratio of the altered fragment(s) to the full-length fragment using a reference fragment from the DNA marker for normalization. The RT-PCR products from selected samples from carriers of the *MSH2* and/or *MSH6* variants were cloned to isolate individual transcripts using the TOPO^®^ TA Cloning^®^ Kit for Sequencing (ThermoFisher Scientific) according to manufacturer’s protocol and sequenced using Sanger sequencing to identify the different altered transcript species. Sequences were aligned using Seqman Pro of the Lasergene software (DNASTAR, Madison, WI, USA). The online ExPASy Translate tool (https://web.expasy.org/translate/, accessed on 3 November 2017) was used to predict the resultant amino acid sequence change from altered mRNA transcripts.

To assess the penetrance of the *MSH6* c.3936_4001+8dup variant on erroneous splicing, RT-PCR was additionally performed between *MSH6* exons 3–10 in a carrier who was also heterozygous for an upstream benign rs1800935C>T single nucleotide polymorphism (SNP) located in exon 3. RT-PCR was conducted using the SuperScript™ IV One-Step RT-PCR System (Appendix A). The RT-PCR products were cloned into the CloneSmart^®^ LCAmp Blunt Cloning Kit (Lucigen, Middleton, WI, USA), transfected into *E. coli*, and plasmids extracted from individual ampicillin-resistant bacterial colonies were Sanger-sequenced with the accompanying vector primers SL1 and SR2, all according to the manufacturer’s recommendations. Sequences from the individual transcripts were aligned using Seqman Pro to determine which genotype of the rs1800935C>T SNP was linked to each normally spliced and erroneously spliced transcript.

### 2.8. Chr2p21-p16 Haplotype Analyses

To assess the maternal contribution of the Chr2p21-p16 region from a female family member who carried both the *MSH2* and *MSH6* variants (III.19) to her son (IV.11) who carried neither, Chromium 10× linked-read whole genome sequencing (10× Genomics, Pleasanton, CA, USA) was performed using leukocyte DNA samples from mother and son. Sequencing libraries were prepared using the Chromium Gel Bead and Library Kit (10× Genomics) and sequencing was performed on an Illumina sequencer (San Diego, CA, USA) with 2 × 151 base-pair paired-end reads at Novogene (Beijing, China). By using the Chromium library preparation, each resulting read pair includes a 16-bp barcode that denotes their high molecular weight (HMW) DNA molecule of origin. Long Ranger v2.2.2 (10× Genomics) commands were used to process the sequencing data. The ‘longranger mkfastq’ command was used to demultiplex and convert the resulting BCL files to FASTQ files. The ‘longranger wgs’ command was used to: (i) align the barcoded reads in the FASTQ files to the human genome reference, (ii) detect single nucleotide variants (SNVs) and small insertion-deletions (indels) by invoking GATK v3.7 [18], and (iii) phase variants using the barcode information associated with each read pair. For our analysis, we used a Long Ranger-compatible version (v2.1.0) of the human GRCh38 reference (downloaded from: http://cf.10xgenomics.com/supp/genome/refdata-GRCh38-2.1.0.tar.gz, accessed on 17 December 2022).

To improve the resolution of the haplotype structure across a homozygous region of the maternal genome, we implemented a targeted Chromium 10× linked-read sequencing method that we previously developed [19]. We designed an assay targeting a 0.3 Mb region (chr2:47307193–47598910) that included the homozygous region. Guide RNAs (gRNAs) were designed using a pipeline where the specificity of the gRNAs is assessed against all possible 20-mer sequences from the GRCh38 genome (Appendix A). Synthetic CRISPR RNAs (crRNAs) and a trans-activating CRISPR RNA (tracrRNA) were purchased from Integrated DNA Technologies (Coralville, IA, USA). Isolation of the 0.3-Mb locus was performed with the “CATCH 100–300kb extr1h sep3h.shflow” workflow on the SageHLS instrument (Sage Science, Beverly, MA, USA). Using whole blood collected in an ACD tube, we first lysed red blood cells, then used only the nucleus-containing cells. Approximately one million intact cells were loaded into the sample well, and cell lysis, genomic DNA extraction, CRISPR-targeted excision, and size selected target elution were performed as instructed by the user guide. Qubit HS (ThermoFisher Scientific, Waltham, MA, USA) and TaqMan qPCR Copy Number (Assay ID: Hs02937482_cn; ThermoFisher Scientific) assays were used to quantify the DNA samples.

Using a 1.25-μL aliquot (0.2 ng) of target-enriched DNA sample from the automated SageHLS (Sage Science) process, we prepared linked-read libraries using the Chromium Gel Bead and Library Kit (10× Genomics). Sequencing was performed on a NextSeq 500 sequencer (Illumina) with 2 × 151 base-pair paired-end reads. The sequencing data were processed as described previously for the whole genome data except that the ‘longranger targeted’ command was used in place of the ‘longranger wgs’ command and with the following parameter additions and adjustments: (i) the ‘target’ parameter specified the CRISPR-targeted region (chr2:47307193–47598910) and (ii) the ‘downsample’ parameter was used to randomly down-sample the data such that 8.4 G of data was used by the pipeline. Down-sampling between 6 G and 10 G was tested in increments of 0.2 G, and 8.4 G of data generated the largest segment of phased variants.

To parse the phased variants in the VCF output files generated by Long Ranger, the gemtools ‘get_phased_basic’ tool from the gemtools python package [20] was used to extract basic data from the phased VCF file to a format conducive to subsequent analysis. Using these files, a custom python script was used to compare the allelic content of each of the mother’s haplotypes to each of the son’s haplotypes and thus determine the maternal inheritance of this region in the son. For this haplotype comparison, we used SNVs that had passed filter with a quality of at least 150.

## 3. Results

### 3.1. Clinical Presentation and Cancer Family History

The proposita (III.20, Figure 1 (blue arrow)), a female of European heritage, was 47 years of age when she presented to a family cancer clinic in Australia in 2012 with concerns about her personal risk of cancer due to a self-reported extensive family history of cancer. She was the youngest of eight siblings and both of her parents and five older siblings had cancer diagnoses consistent with LS. The proposita had undergone a full hysterectomy and salpingo-oophorectomy at age 43 years for clinical indications but review of the pathology report showed no evidence of malignancy. A pedigree, “Family C”, was constructed based on the detailed information the proposita provided, which comprised four generations, with 13 branches descending from the proposita’s cancer-affected grandmother of German heritage in the first generation (I.3, Figure 1). The pedigree met Amsterdam I criteria for Lynch syndrome [21] with a remarkable history of cancer spanning three generations (I-III) in 11 of the 13 branches, in particular young age-of-onset CRC and EC and other diagnoses consistent with LS. There were no cancer diagnoses in generation IV at last contact; however, these members were all under 40 years of age, younger than the average age of onset of first LS-associated CRC (<44 years).

In 1995, the proposita’s deceased eldest brother, herein “the proband” (III.13, Figure 1 (black arrow)), had presented with synchronous colon (cecal, stage IV) and rectal adenocarcinoma (stage II) at age 55 years. A biopsy of the rectal tumor was tested for MMR deficiency and had demonstrated MSI-high with dual loss of MSH2 and MSH6 and retention of MLH1 and PMS2 by immunohistochemistry (IHC), raising a clinical suspicion of LS. The proband had undergone germline screening of *MSH2* and *MSH6* by Sanger sequencing and multiplex-ligation probe amplification (MLPA) gene dosage analyses, which revealed two genetic variants, *MSH2* c.2006G>T (p.Gly699Val) and *MSH6* intron 9, IVS9+8_+9ins73, now reported as *MSH6* c.3936_4001+8dup (intronic). Both variants were classified as VUS at that time. A cousin in branch 2 (III.2, Figure 1) who had developed breast cancer at age 46 years and endometrial cancer at age 60 years subsequently underwent genetic testing of the *BRCA1/2* and MMR genes in 2010 in which the *MSH2* c.2006G>T variant was identified and again reported as a VUS. Neither *MSH2* c.2006G>T nor *MSH6* c.3936_4001+8dup variant was reported in population-based databases. Thus, at the time the proposita presented to the family cancer clinic, the limited genetic testing that had been conducted in her cancer-affected relatives had yielded uninformative results (VUSs).

To determine if the proposita was a carrier of either variant or another high-penetrance LS-causing pathogenic variant, we performed complete screening of the *MLH1*, *MSH2*, and *MSH6* MMR genes by Sanger sequencing across all exons and exon-intron boundaries as well as dosage analysis by MLPA of *MLH1*, *MSH2*, *MSH6* and *EPCAM* in peripheral blood leukocyte (PBL) DNA. The proposita was a heterozygous carrier of both the *MSH2* c.2006G>T (Figure 2A) and *MSH6* c.3936_4001+8dup (Figure 2B) variants. No other variants were identified and MLPA showed all four genes were intact. To resolve the *MSH6* c.3936_4001+8dup variant, clonal sequencing of the two *MSH6* alleles was performed.

The variant comprised a 74 bp tandem duplication consisting of a segment of exon 9 and the first 8 bp of intron 9, thereby encompassing the donor splice site at the exon-intron boundary (Figure 2B, Appendix A). The proposita was recommended to undergo surveillance for LS-associated cancers based on her family history whilst definitive evidence regarding the potential pathogenicity of the two VUS was sought. During subsequent endoscopic screening a colonic polyp was identified at age 50 years, and multiple gastric polyps and a large duodenal adenoma with high-grade dysplasia were identified at age 55 years, which were endoscopically resected.

### 3.2. Molecular Analyses of Archived Adenocarcinoma

To determine the role of the *MSH2* and *MSH6* variants in the molecular pathogenesis of cancer, we retrieved the FFPE tissue blocks and accompanying pathology reports that were available for the adenocarcinomas from two of the proposita’s deceased brothers. These included the rectal tumor from proband (III.13) and the stage II transverse colon tumor from III.14 diagnosed at age 53 years, both of which had demonstrated MSI and dual loss of MSH2 and MSH6 immunohistochemical (IHC) expression. We performed targeted testing of DNA from the adjacent normal colorectal mucosa (NCM) and tumor tissues to determine carrier status of the *MSH2* and *MSH6* variants and seek loss-of-heterozygosity (LOH) as a potential “second hit” in tumorigenesis. Given the proband’s genetic testing in 1995 predated the discovery of 3′ *EPCAM* deletions in association with *MSH2* CpG island promoter methylation and IHC loss of expression [3], we also tested for the presence of *MSH2* promoter methylation by CpG pyrosequencing in the two tumors and adjacent NCM tissues.

We confirmed that the proband (III.13) carried both the *MSH2* c.2006G>T and the *MSH6* c.3936_4001+8dup variants (Figure 2C,D). In the rectal tumor, only the *MSH2* c.2006T allele was detected, indicating complete LOH of the wild-type c.2006G allele (Figure 2C). By contrast, the wild-type *MSH6* allele was retained in the tumor and the variant *MSH6* c.3936_4001+8dup allele was reduced (Figure 2D). This finding was consistent with MSH2-deficiency having driven tumorigenesis in this rectal cancer (germline c.2006T allele serving as the “first-hit” and somatic LOH of the wild-type c.2006G allele as the “second-hit”) with secondary loss of MSH6 expression as a consequence of its dependency on intact MSH2 as opposed to any involvement by the *MSH6* c.3936_4001+8dup variant. The other brother, III.14, also carried both variants. The wild-type *MSH2* c.2006G allele was diminished as compared to the NCM but did not reach the criteria for LOH (Appendix A). Both the tumor and adjacent NCM tissues from both brothers were negative for methylation at the *MSH2* promoter, consistent with intact *EPCAM* (Appendix A). Collectively, these findings were consistent with the *MSH2* c.2006G>T variant predisposing to tumorigenesis in both MSH2-deficient cancers, most definitively in the proband’s rectal cancer, whilst arguing against a role for *EPCAM* deletion and the *MSH6* c.3936_4001+8dup variant in their development.

### 3.3. Extended Pedigree Analysis for Carriage of the MSH2 c.2006G>T and MSH6 c.3936_4001+8dup (Intronic) Variants and Co-Segregation with Major LS Cancers

To seek evidence of a differential role for the *MSH2* and *MSH6* variants in the cancer history in Family C, targeted genetic testing for carrier status was performed in consenting family members from the extended pedigree followed by co-segregation analyses of each variant with disease status. Fifty-nine living family members spanning three generations (II–IV) from 11 of the 13 branches joined the study and provided their personal and family history of pre/neoplasia, treatment, pathology reports, and biological samples for DNA and RNA analyses. Carrier status for both variants was determined for all 59 persons. Through pedigree analyses, we were additionally able to infer obligate carriers of each variant among family members for whom no sample was available (Figure 1, labeled OC). *MSH2* c.2006G>T was identified in multiple cancer-affected members of generation III from seven of the 11 branches tested. This variant was traced back through obligate carriage to seven deceased siblings in generation II with diagnoses of LS-associated cancers. In branch 2, several carriers of the benign *MSH2* c.2006-6T>C variant (rs2303428, minor allele frequency, MAF 9.6%) [10], were identified among both carriers and non-carriers of *MSH2* 2006G>T, hence the two *MSH2* variants were non-segregating.

The *MSH6* c.3936_4001+8dup (intronic) variant was limited to branch 5 within the extended Family C pedigree in which it co-occurred with the *MSH2* c.2006G>T variant (Figure 1). Within branch 5, of the six siblings within generation III tested, four carried both the *MSH2* and *MSH6* variants (III.13, III.14, III.19, and III.20), and two living brothers (III.15 and III.18) carried the *MSH6* variant alone. III.18 had developed a sebaceous carcinoma of the skin at the age of 52 years. Both III.15 and III.18 had developed multiple polyps beginning in their fifties, which were endoscopically resected. The report of a screening colonoscopy performed on subject III.18 at age 55 years found three flat polyps of 3-4 mm diameter in the sigmoid colon and cecum. Diagnoses of additional colorectal polyps developed by these two carriers of the *MSH6* variant alone during the course of this study are detailed below. In generation IV, six of the nine members tested carried either the *MSH2* variant or the *MSH6* variant but none carried both (Figure 1).

*MSH2* and *MSH6* are co-localized on Chr2p21-p16, separated by the *KCNK12* gene. The *MSH2* c.2006G>T and *MSH6* c.3936_4001+8dup variants segregated independently in branch 5 of the family, indicating they are located on separate homologues of chromosome 2. On this basis, we inferred that the *MSH6* c.3936_4001+8dup (intronic) variant was introduced into the larger *MSH2*-LS family by the proposita’s mother (II.9, Figure 1) who had developed EC at age 58 years and skin cancers at ages 59 and 63 years. The mother’s pedigree was constructed based on data provided by the proposita to determine if there was a history of cancer consistent with *MSH6*-LS (Figure 1). Cancer diagnoses were identified on the mother’s maternal side, including a colon cancer at age 66 years in an uncle, breast cancer age 37 years in a cousin, and another cancer diagnosis of unknown site in another cousin. No members of this family have undergone genetic testing and none consented to join this study.

Co-segregation analysis was performed for the *MSH2* c.2006G>T and *MSH6* c.3936_4001+8dup (Intronic) variants independently for CRC and EC, the two most frequent LS-associated cancers. Briefly, this approach is an age-dependent model, referred to as the survival model, that considers an individual unaffected until their age of diagnosis. Thus, the survival model can provide evidence against pathogenicity if an individual presents very late-onset disease. We observed a co-segregation LOD score of 4.99 and Bayes factor (B) score of 98,044 for *MSH2* c.2006G>T using the cancer incidence rate of non-Finnish Europeans during 2008–2012. This provides very strong evidence for co-segregation of the *MSH2* c.2006G>T variant with CRC and EC based on the American Medical Genetics and Genomics (ACMG) and the Association for Molecular Pathology (AMP) conversion method [22]. For *MSH6* c.3936_4001+8dup (Intronic), the co-segregation analysis was limited to branch 5 within the extended pedigree of Family C (Figure 1). For CRC and EC, we observed a co-segregation LOD score of −3.1 and a BF score of 7.77 × 10^−4^, suggesting lack of co-segregation. Since the strong adherence to colonoscopic surveillance and hysterectomies among members of branch 5 may have reduced the risk of developing cancer, a second analysis was performed in which colorectal polyps were included as a disease outcome. We observed a co-segregation LOD score of 0.235 and a BF score of 1.72 for *MSH6* c.3936_4001+8dup (Intronic), boosting this variant into the uncertain significance range (BF 0.48–2.08).

### 3.4. Risk for First Occurrence of Neoplasia in Carriers of the MSH2 and MSH6 Variants

To assess the risk for cancer onset associated with carriage of either *MSH2* c.2006G>T or *MSH6* c.3936_4001+8dup (Intronic) or both, we plotted Kaplan–Meier curves of (i) first onset of any cancer, and (ii) first onset of any cancer or colorectal polyp(s) pooled (pre/neoplasia) among segregated groups of confirmed/obligate carriers of *MSH2* c.2006G>T alone (*n* = 26), *MSH6* c.3936_4001+8dup (intronic) alone (*n* = 7), both variants (*n* = 4), and non-carriers of either variant (*n* = 27) (Figure 3).

There was a significant difference (*p* = 0.009) in the age of onset between the groups of carriers of *MSH2* c.2006G>T alone (mean age 49.7 years with 95% confidence interval [CI], 45.2–54.3), *MSH6* c.3936_4001+8dup (intronic) alone (mean age 56.0 years with 95% CI, 51.5–60.5), both variants (mean age 54.0 years with 95% CI, 52.0–56.0), and non-carriers (mean age 63.0 with 95% CI, 59.9–66.1) (Figure 3A). When polyps were included with first cancer occurrence, the cumulative risk for pre/neoplasia onset was also significant (*P*=0.007) between the groups of carriers of *MSH2* c.2006G>T alone (mean age 49.5 years with 95% CI, 45.1–53.9), *MSH6* c.3936_4001+8dup (intronic) alone (mean age 54.7 years with 95% CI, 51.2–58.1), both variants (mean age 52.5 years with 95% CI, 51.4–57.1), and non-carriers (mean age 61.5 years with 95% CI, 57.3-65.8) (Figure 3B). No additive risk for onset of cancer or pre/neoplasia was observed among dual-variant carriers over and above that of either variant alone, although the number of subjects included in this group was small and, furthermore, both female dual-variant carriers had undergone prior complete hysterectomies for clinical reasons other than malignancy. The age at cancer onset among carriers of *MSH6* c.3936_4001+8dup (intronic) alone versus non-carriers of either MMR variant did not reach statistical significance (*p* = 0.08, Figure 3C). However, when preneoplasia and first cancer were pooled, the risk for onset of pre/neoplasia was significantly increased among carriers of *MSH6* c.3936_4001+8dup (intronic) alone, (*p* = 0.034, Figure 3D).

### 3.5. IHC Analyses of Adenomas in Carriers of MSH6 c.3936_4001+8dup (Intronic) Alone

Among identified or obligate carriers of the *MSH6* c.3936_4001+8dup (intronic) variant alone, no cancer tissue samples were available to resolve the pathological role for this variant. However, prior studies have shown that approximately three-quarters of LS-associated colorectal adenomas demonstrate IHC loss of expression of the relevant MMR protein [23,24,25]. Two of the proposita’s living brothers, III.15 and III.18, both of whom had developed multiple colonic polyps in their fifties and above, had developed additional polyps during the course of this study, which had been endoscopically resected during one of their regular surveillance colonoscopies. We retrieved the FFPE tissues from these recent polyps. Histopathologic review by a GI pathologist showed these comprised one 3 mm colonic tubular adenoma from III.15 at age 66 years and one 2–3 mm hyperplastic polyp and one 2–3 mm colonic tubular adenoma from III.18 at ages 57 and 59 years, respectively. The tubular adenomas were considered early, low-grade preneoplasia (they lacked any villous or high-grade dysplasic features associated with more advanced adenomas). IHC of all four MMR proteins was performed on these preneoplasia and adjacent normal tissues and all were intact for each of the MMR proteins (Appendix A). No focal losses of MSH6 expression were observed in any of the three polyps. Unfortunately, insufficient specimen was available for MSI testing.

### 3.6. Functional Analyses of the MSH2 in RNA of Carriers Show MSH2 c.2006G>T Has Differential Effects on Aberrant Splicing in Different Tissue Samples

*MSH2* c.2006G>T encodes the missense variant p.(Gly669Val), which in 2012, was predicted by the “Pathogenic-or-not” PON-MMR in silico prediction tool to be loss-of-function since the glycine at p.669 is highly evolutionarily conserved and the valine that replaces it has distinct physicochemical properties [26]. In addition, c.2006G>T is located at the canonical +1 position of the splice acceptor site of exon 13. In 2015, *MSH2* c.2006G>T was shown to cause partial out-of-frame skipping of exon 13 leading to a premature termination codon (PTC) in minigene assays and loss-of-function [12]. In 2017, partial exon 13-skipping was observed as well as the full-length missense p.Gly699Val-encoding transcript in white blood cells and cultured lymphoblasts from three independent carriers in Europe [13]. On the basis of these findings, and additional collective evidence from clinical and molecular tumor characteristics, *MSH2* c.2006G>T was reclassified as a class 5 PV in 2017 [13].

To assess the functional effect of *MSH2* c.2006G>T on mRNA splicing in members of Family C, we first performed reverse-transcriptase (RT) PCR between *MSH2* exons 12, 13, and 14 in peripheral blood mononuclear cells (PBMC) from the proposita (III.20). Gel electrophoresis revealed a full-length and a shorter RT-PCR product for the exon 12–14 cDNA amplicons (Figure 4A). Direct sequencing of the exon 12–14 cDNA amplicons showed a frameshift in the transcript beginning from the c.2007 position with the asynchrony produced by overlapping exon 13 and 14 sequences (Figure 4B). No trace of the variant T allele at the c.2006 position was observed above the background-noise levels in the sequence electropherograms. This finding was consistent with complete skipping of exon 13 in transcripts derived from the variant c.2006T allele in PBMCs. In 2013, we shared these data with the genetic testing company that had previously identified the *MSH2* c.2006G>T variant in the cancer-affected cousin (III.2, Figure 1). Following internal review of the data, the genetic testing company considered it insufficient evidence for reclassification of the variant from VUS.

In the whole blood and saliva mRNA from subject III.19 and saliva mRNA samples from additional carriers of the *MSH2* c.2006G>T variant from different branches of Family C, the shorter fragment representing the exon 13-skipped cDNA amplicon was fainter than the full-sized fragment in electrophoresis gels (Appendix A). Using clonal sequencing of selected exon 12–14 RT-PCR products to isolate and sequence individual transcripts from selected samples (III.3, III.11, III.19, III.20), we identified three distinct *MSH2* transcripts: the normal transcript derived from the wild-type allele and two derived from the variant allele; the exon 13-skipped transcript and the full-length missense c.2006T, p.Gly669Val-encoding transcript (Figure 4C). Among the clones sequenced, the missense transcript was the most abundant of the variant transcripts in saliva, whereas the exon 13-skipped transcript was the most abundant variant transcript in whole blood. To minimize potential biases incurred by variations in batch processing or the duration of sample storage, we simultaneously collected and processed fresh samples of PBMC, hair follicles, and saliva from a single carrier of the *MSH2* c.2006G>T variant, cancer-affected subject III.3 from branch 2 of Family C (Figure 1). Using both densitometry of the transcripts after size-fractionation by gel electrophoresis Figure 4D) as well as random clonal sequencing of the RT-PCR amplicons (Figure 4E), we found the relative abundance of the exon 13-skipped and missense transcripts differed between the three tissue-types from this subject. The same *MSH2* exon 12–14 RT-PCR in saliva from non-carrier members of Family C did not show any abnormal fragments. Similarly, no abnormal sized transcripts were identified by RT-PCR in PBMCs, saliva, and hair follicles from a healthy control subject and sequencing of the amplicons showed the wild-type full-length transcripts (Appendix A).

### 3.7. Functional Analyses of MSH6 in RNA of Carriers Show MSH6 c.3936_4001+8dup (Intronic) Causes Out-of-Frame Splicing Errors

The *MSH6* c.3936_4001+8dup (Intronic) variant involves a duplication of the exon 9 donor splice site within intron 9 (Figure 2). To determine in patient samples if this variant had a detrimental impact on mRNA splicing, we performed RT-PCR between *MSH6* exons 8–10 on RNA from leukocytes and/or saliva from several carriers of the *MSH6* c.3936_4001+8dup (intronic) variant, followed by densitometry of size-fractionated amplicons to determine the relative amounts of each altered fragment. In addition, we cloned selected RT-PCR products to isolate and sequence individual transcripts. In some carriers, we found the *MSH6* c.3936_4001+8dup (intronic) variant associated with a single aberrant transcript visualized on electrophoresis gels as a shortened fragment at a ratio of approximately 50%:50% transcript levels with the expected normal-length wild-type fragment (as illustrated for III.18 in Figure 5A, right). In other carriers of *MSH6* c.3936_4001+8dup (intronic), two distinct transcripts were observed; the shortened fragment plus another lengthened fragment at slightly higher levels, as detected by densitometry of the respective bands in electrophoresis gels (as illustrated for III.15 and III.19 in Figure 5A).

Clonal sequencing of the exon 8–10 RT-PCR amplicons from subjects III.15 (saliva) and III.19 (white blood cells and saliva) showed the wild-type transcript plus two distinct out-of-frame transcripts that arose from altered splicing at different splice sites. One of these was the shortened transcript, which resulted from skipping of exon 9 (Figure 5B). The adjoining of exon 10 to exon 8 resulted in the deletion of amino acid residues 1268–1360 (encoded by the full exons 9 and 10) with a novel sequence of five final amino acid residues introduced by the frame-shifted exon 10 prior to the introduction of a PTC (Figure 5B). The protein encoded by this exon 9-skipped transcript was thus predicted to result in loss-of-function via deletion of the entire MSH2-interaction domain and truncation of the ATPase domain within the translated protein product. The second, lengthened aberrant transcript included the normal exon 9 followed by the first 8 bp of intron 9 immediately preceding the duplication plus the duplicated region of (partial) exon 9 followed by exon 10, having omitted the final 8 bp of the duplicated intron 9 sequence present in the genomic DNA (Figure 5B). Thus, in this altered transcript the normal exon 9 donor splice site was skipped over (hence the inclusion of 8 bp of intron 9) but the duplicated exon 9 donor splice site introduced further downstream was utilized and spliced directly onto exon 10. This aberrant transcript was predicted to result in the introduction of a PTC starting at position +2 in the retained 8 bp segment of intron 9 (Figure 5B). This transcript was predicted to result in loss of amino acid residues 1134–1360 at the C-terminus and, hence truncation of the MSH2-interaction and the ATPase domains with loss-of-function effect. Loss of the C-terminus has been shown to severely compromise the function of the MutS homologue in *E. coli* [27]. Sequencing of *MSH6* exon 8–10 RT-PCR amplicons in non-carriers and a healthy control showed the wild-type transcript (Appendix A).

Next, to determine if the variant *MSH6* c.3936_4001+8dup (Intronic) allele was fully penetrant (produced no normal transcripts), we used an upstream SNP (rs1800935C>T) within *MSH6* exon 3 for which individual III.19 (proband/proposita’s sister) was also heterozygous, as a tag-SNP to track which genetic allele the normal and erroneously spliced transcripts were derived from. Linked-read sequencing of genomic DNA in III.19 (see below) had shown the wildtype *MSH6* haplotype was linked to the C genotype at rs1800935 and the c.3936_4001+8dup (Intronic) variant haplotype was linked to the T genotype at rs1800935. Using mRNA derived from a fresh sample of PBL, RT-PCR was performed from exon 3 to exon 10 to include both the SNP and normal/splice variant transcripts at exons 8-10 within the amplified products (Figure 6A), cloned and sequenced in both directions with vector primers. Sequencing revealed that all normal transcripts contained only the C genotype at rs1800935 exon 3 (Figure 6B). All transcripts containing the T genotype at rs1800935 were erroneously spliced (Figure 6C).

These included the same two frame-shift transcripts observed in our prior RT-PCR analyses across exons 8–10. Interestingly, the transcript with partial inclusion of the duplication was predominant over the exon 9-skipped transcript at a ratio of 7:1. The findings of varying proportions of the two altered splice transcripts across the different samples (PBL, PBMC, saliva) among carriers in the family suggests the mechanism of altered splicing associated with the *MSH6* c.3936_4001+8dup (Intronic) variant is variable. Most importantly, these analyses found no normal transcripts derived from the haplotype bearing the *MSH6* c.3936_4001+8dup (Intronic) variant, consistent with full penetrance for erroneous splicing.

Based on these functional analyses, and consistent with the MMR gene variant classification criteria assembled by the InSiGHT variant interpretation committee [28], we conclude that the novel *MSH6* c.3936_4001+8dup (Intronic) variant is a fully penetrant pathogenic splice variant associated with loss-of-function frameshift transcripts that encode premature truncation of the MSH6 protein. Therefore, we proffer that this variant is likely pathogenic, representing an independent cause for LS in branch 5 of Family C, in addition to the *MSH2* c.2006G>T PV.

### 3.8. Extended Screening for Germline Mutations in Cancer-Affected Members Who Did Not Carry Either MSH2 or MSH6 Variant

No consenting members of branch 8 carried either the *MSH2* or *MSH6* variant, yet three members had prior cancer diagnoses (Figure 1), including diagnoses of bilateral ovarian cancer in a female aged 65 years in II.9, non-Hodgkin’s lymphoma (NHL) at age 52 years in one of her daughters (III.24), and skin basal cell carcinoma at age 52 and 53 years in another daughter (III.26). Exome sequencing was performed in subject II.9 to identify any other potential cancer-associated variants. A multitude of VUS in cancer-relevant genes was found but the only variant with a clear association with elevated cancer risk was the heterozygous loss-of-function frameshift variant *MC1R* c.86dupA (p.Asn29Lysfs*14; rs796296176), which has been associated with increased risk for skin cancers [29] but not ovarian cancer. The heterozygous missense variant *PTPRJ* c.827A>C (p.Gln276Pro, reference sequence NM_002843.3, SNP rs1566734) was also identified in II.9 and in her NHL-affected daughter (III.24). This variant has been implicated in NHL [30] but was found not to be associated with elevated risk for solid tumors [31].

### 3.9. Genetic Counseling and Cancer Gene Panel Testing Provides Diagnostic Genetic Test Results and Identifies Additional Cancer-Risk Genes in the Extended Pedigree

Prompted by the reclassification of *MSH2* c.2006G>T as a class 5 PV in 2017 [13], we approached and facilitated all consented family members to undergo genetic counseling and diagnostic molecular genetic testing in a CLIA-approved facility (genetics service provider, Invitae). Forty-one of the 59 consented family members opted to undergo genetic counseling and gene panel testing using the Invitae 134 cancer gene panel test (details provided in Appendix A). Formal genetic test results for *MSH2* c.2006G>T and *MSH6* c.3936_4001+8dup (Intronic) were consistent with in-house research results for all 41 subjects tested in both. The genetic test reports conveyed the *MSH2* c.2006G>T as a PV, whereas the *MSH6* c.3936_4001+8dup (intronic) variant was reported as a VUS in the absence of corroborating functional or clinical data. Findings from the novel *MSH6* c.3936_4001+8dup (intronic) variant were submitted by Invitae to ClinVar (www.ncbi.nlm.nih.gov/clinvar/variation/663057/, first submission on 14 August 2019).

Interestingly, in addition to the two MMR gene variants, cancer gene panel testing revealed two additional cancer-associated PVs in family members within branches that did not carry either MMR variant. These included the skin cancer-associated *MC1R* c.85dupA (p.Asn29Lysfs*14) variant in multiple subjects in branches 6, 8, 10, and 11 in which one carrier (subject III.26) had a history of skin basal cell carcinoma (Figure 1). *NBN* c.657_661del (p.Lys219Asnfs*16), an Eastern European founder loss-of-function frameshift variant associated with increased risk for breast, prostate, colorectal, and additional cancers [32,33,34], was identified in three siblings in branch 10, who were unaffected by cancer at the close of study (Figure 1). Thus, although the living members of branches 6, 8, 10, and 11 in Family C did not have LS, carriers of the *NBN* and/or *MC1R* variants were counseled about the current state of knowledge of associated cancer risks and strategies for risk management.

### 3.10. An Instance of Non-Transmission of Either MSH2 or MSH6 Variant to Offspring

Given the *MSH2* c.2006G>T and *MSH6* c.3936_4001+8dup (intronic) variants were non-segregating and hence located on separate Chr2p21-p16 homologues, for digenic carriers of both variants in generation III of branch 5, the risk of transmission of either one or other variant was predicted to be ~100%. Thus, the offspring of dual-variant carriers were at near 100% risk of inheriting either *MSH2*-LS or *MSH6*-LS causing variants. Nevertheless, dual-variant carrier III.19 (proband/proposita’s sister) did not transmit either variant to her eldest son (IV.11), although her youngest son (IV.12) did inherit the *MSH6* variant (Figure 1). Various hypotheses could explain such a finding of non-transmission, including: (1) a sampling error; (2) a de novo deletion encompassing and thereby removing either *MSH2* or *MSH6* variant in the son (IV.11), which would incur another cause for LS; (3) complete or segmental paternal uniparental disomy (pUPD) for chromosome 2, or Chr2p21-p16, such that IV.11 had inherited both copies of the Chr2p21-p16 region from his father with no maternal contribution, which has previously been described in healthy subjects [35,36,37]; or (4) a recombination event between the locations of the *trans MSH2* c.2006G>T and *MSH6* c.3936_4001+8dup (Intronic) variants in the maternal germline such that the son inherited the recombined wild-type allele from his mother.

To rule out a sampling error and detect any potential deletion, both mother (III.19) and son (IV.11), underwent complete genetic testing for LS by NGS and array comparative genomic hybridization (aCGH) at an independent CLIA-approved laboratory using fresh blood samples. This re-testing confirmed the mother (III.19) was a carrier of both *MSH2* c.2006G>T and *MSH6* c.3936_4001+8dup (Intronic) variants and the son (IV.11) carried neither variant and no deletion was detected.

The father of IV.11 was deceased; therefore, paternal contribution to the Chr2p21-p16 region could not be assessed directly. Therefore, to assess the genetic contribution of maternal alleles and any evidence for recombination between alleles, we performed Chromium 10× linked-read genome sequencing (lr-GS) followed by haplotype reconstruction on high molecular weight leukocyte DNA from both mother (III.19) and son (IV.11). Lr-WGS was able to generate phased haplotypes for the son across the entire Chr2p21-p16 region of interest (Appendix A); however, his mother contained an approximately 200 kb stretch of near complete homozygosity (chr2:47357917–47552885) such that the adjacent regions could not be confidently phased with respect to one another. Using targeted linked-read sequencing with CRISPR guides, we were able to span the region of homozygosity to generate complete haplotypes across the region for the mother (Figure 7; Appendix A). Notably, one of the son’s haplotypes (haplotype 1) comprised SNP alleles across the Chr2p21-p16 region that were consistent with inheritance from his mother (Figure 7), thereby ruling out pUPD in the Chr2p21-p16 region since. Interestingly, we did find evidence for a recombination between the maternal alleles in the son with the breakpoint lying between genomic positions chr2:47476367 and chr2:47552885. The recombination breakpoint is located within the intervening region between the two *MSH2* c.2006G>T and *MSH6* c.3936_4001+8dup (intronic) variants. This would have given rise to a wild-type haplotype across the two MMR genes, thereby accounting for the non-transmission of either variant from mother to son.

## 4. Discussion

Herein, we describe a multibranched family with *MSH2*-LS caused by the c.2006G>T (p.Gly669Val) pathogenic variant in seven of the 11 branches screened in which one branch has an additional, independently segregating, novel *MSH6* c.3936_4001+8dup (Intronic) variant. Both variants are out-of-frame splice variants resulting in the introduction of a PTC, consistent with pathogenicity. Even though we identified two pathogenic MMR variants *in trans* within branch 5 of Family C, they do not have CMMRD [1] since the variants affect different MMR genes. Rather, dual carriers of these two variants have digenic LS. Previous cases with germline variants in two MMR genes have been described but they were either located in cis [38] or one or both of the variants was predicted to be a VUS or benign [39,40,41]. The finding of two independent LS-causing variants within these two different MMR genes in one branch of Family C has strong implications for cancer risk stratification given the overall higher risks for cancer in carriers of pathogenic variants of *MSH2* than *MSH6*, clinical surveillance, and genetic counseling for their respective risks of cancer and inheritance.

At the time the *MSH2* c.2006G>T (p.Gly669Val) and *MSH6* c.3936_4001+8dup (Intronic) variants were first identified in the proband, and thereafter when the proposita came forward, they were both classified as VUS. We used a multipronged approach to assess the contribution of each MMR variant to cancer development in this family. In terms of clinicopathologic findings, we found strong, correlative evidence of a role for the *MSH2* c.2006G>T variant in cancer development. This is in agreement with its classification as a class 5 PV in 2017, supported by a combination of functional, co-segregation, and clinicopathological evidence in a multifactorial likelihood analysis [13]. Our observations of MSI, loss of MSH2 immuno-expression, and LOH of the wild-type allele in the proband’s rectal adenocarcinoma and similar findings in the tumor of his deceased brother were consistent with the *MSH2* c.2006G>T variant serving as the predisposing “first hit” in cancer development. This variant also co-segregated with the major LS-associated cancers (CRC and EC) in Family C and a significant cumulative risk for cancer onset by age was observed in carriers, comparable to that for *MSH2*-LS in the literature [5,7]. Interestingly, carriers of both variants did not have any additive risk above that for the *MSH2* variant alone for cancer onset although the numbers were too few for an accurate assessment of this.

The *MSH2* c.2006G>T variant gave rise to full or partial erroneous splicing that results in out-of-frame skipping of exon 13 in the RNA of carriers within Family C, consistent with the findings by others [13]. In other studies, both complete and partial splicing had been observed in whole blood and lymphoblastoid cells lines from different carriers, raising the question of whether altered splicing was the sole mechanism for MSH2 inactivation [12,13]. Further to this, we found variable levels of relative abundance of the exon 13-skipped transcript compared to the normally spliced missense c.2006T (p.Gly669Val-encoding) transcript in different tissues within the same subject, despite simultaneous collection and processing of the samples. The exon 13-skipped transcript levels varied from 9% in saliva to 59% in PBMC. We considered possible explanations for these differences: *MSH2* c.2006G>T may have differential tissue-specific effects on the penetrance of splicing activity. PBMCs have been shown to express more splicing factors compared to other blood components such as granulocytes, which may be required for transcription and protein synthesis in physiological responses of lymphocytes and monocytes to pathogens [42]. Expression of splicing factors may also be higher in PBMCs than in hair follicles and saliva, resulting in higher penetrance of aberrant splicing conferred by splice variants in PBMCs [43]. Alternatively, the rate of cell proliferation or of nonsense-mediated mRNA decay (which would affect the exon 13-skipped transcript only) may differ between cell types. In an vitro functional assay based on complementation of MMR activity in MMR-deficient cell lines, the missense p.Gly669Val protein did indeed abrogate MMR activity [13]. The missense variant replaces a highly conserved glycine at codon 669 located within the core of the folded protein, with the more hydrophobic and bigger valine that may not fit into the core. Therefore, whether the exon 13-skipped or missense transcript is most abundant in any particular tissue, MSH2 function overall would be impaired. The wide spectrum of cancers observed (CRC, EC, ovarian, renal, brain, breast, and upper GI) within the extended “Family C” are entirely consistent with high penetrance of the *MSH2* c.2006G>T PV across multiple tumor tissues-of-origin.

For the *MSH6* c.3936_4001+8dup (intronic) variant, clinicopathologic evidence of a pathogenicity was equivocal. The occurrences of endometrial and skin cancers in the proposita’s deceased mother (II.9, obligate carrier) and skin sebaceous cancers and/or multiple colorectal and upper GI adenomas in confirmed carriers of the *MSH6* variant alone, all above 50 years of age, are consistent with the *MSH6*-LS phenotype. In a recent prospective study, high risks for endometrial cancer were observed in females and only moderately increased risk for CRC in both sexes, with onset from 50 years of age [6]. However, the full extent of the pathogenic role for the *MSH6* variant in branch 5 of Family C may have been obscured in generation III somewhat by the higher-penetrance *MSH2* variant in carriers of both MMR variants, as well as the cancer preventive measures taken by family members (in both generations III and IV) on account of their extensive cancer family history. The proposita and her younger sister, both dual-variant carriers, had both undergone prior complete hysterectomies, thereby precluding subsequent endometrial and ovarian cancers for which they were at high risk. All living members of this branch of the family have also been adherent with regular colonoscopic surveillance recommendations, which has provided the opportunity for early detection and endoscopic resection of polyps. Taking these cancer-preventive measures into account, we assessed the cumulative risk of first onset of pre/neoplasia (cancer and polyps combined) by age in carriers of the *MSH6* variant alone over non-carriers of either MMR variant and did observe a significantly increased risk. IHC analysis of MMR expression in three polyps/adenomas excised from two carriers of the *MSH6* variant alone showed intact MSH6 expression, which was unsurprising, given histopathologic examination of the three polyps tested showed two were early, low-grade, and small tubular adenomas and one was a small hyperplastic polyp. Prior studies have shown that a proportion of LS-associated colorectal polyps display IHC loss of the relevant MMR protein(s), and MSI, but this is more prevalent in conventional adenomas with advanced features including villous component, high-grade dysplasia, or larger size (77% and above), than in small, low-grade, early adenomas (up to 38%) [25,44,45]. These studies also found LS-associated hyperplastic polyps were intact for MMR protein expression. Therefore, the finding of intact MSH6 expression in these polyps does not refute the possibility they were associated with *MSH6*-LS; however, they could also be sporadic occurrences given they were identified above 60 years of age. Unfortunately, none of the younger age-of-onset polyp or skin cancer tissues were available from these two family members in generation III and neither was the mother’s uterine cancer. Therefore, no direct link could be made between the *MSH6* c.3936_4001+8dup (intronic) variant and MSH6-deficiency in any cancer or preneoplasm. Carriers of the *MSH6* c.3936_4001+8dup (intronic) variant alone, in generation IV, are currently under 40 years of age, younger than the age of onset typical for *MSH6*-associated LS cancers [6].

For the *MSH6* c.3936_4001+8dup (intronic) variant, the strongest evidence for pathogenicity provided in this study was the demonstrated detrimental impact on RNA splicing observed in biospecimens from several carriers. This variant was associated with two distinct out-of-frame PTC splice transcripts at different loci within the protein. Interestingly, the relative levels of these two distinct altered splice transcripts were variable across the samples provided by the different family members who were carriers; however, both transcripts were predicted to result in loss of protein function. The shortened exon 9-skipped transcript went out-of-frame at the end of exon 8 resulting in the introduction of five novel residues prior to truncation (with omission of amino acids encoded by exons 9 and 10). The protein produced by the lengthened splice-variant transcript was predicted to truncate at the end of penultimate exon 9. Classification guidelines for the interpretation of (potential) splice variants state that a variant is considered pathogenic (class 5) when it shows aberrant splice products in patient RNA analysis that result in the introduction of a PTC or an in-frame deletion that disrupts known functional domains and produces no normal transcripts [46]. These guidelines have been adopted for the classification of MMR gene variants for the diagnosis of LS [47]. For MSH6, the MSH2-interaction and ATPase domains located at the C-terminus have been shown to be essential for protein function and these domains are truncated or omitted by the two erroneous splice transcripts found in carriers of the *MSH6* c.3936_4001+8dup (intronic) variant. Furthermore, using an upstream tag-SNP to trace the allelic origin of the normal and erroneously spliced transcripts, we found no evidence that the haplotype bearing the *MSH6* c.3936_4001+8dup (intronic) variant produced normal transcripts. It is possible that the method we used may have missed a low fraction of any such normal transcripts from the *MSH6* c.3936_4001+8dup (Intronic) variant allele; however, it remains possible that splice variants that are not fully penetrant do nevertheless elevate the risk for cancer, albeit perhaps not to the same extent as other types of deleterious mutations. On this basis, we conclude that the *MSH6* c.3936_4001+8dup (Intronic) variant should be classified, at the very least, as “likely pathogenic”.

Reclassification of the *MSH2* c.2006G>T variant from VUS to class 5 during the 10 years of follow-up in this family study had a significant impact on its members. This enabled family members to undergo CLIA-approved diagnostic genetic testing and accurate genetic counseling. However, the *MSH6* c.3936_4001+8dup (Intronic) variant is currently reported as a VUS by the CLIA-approved genetic testing facility. The evidence provided in our study is supportive of the reclassification of this variant to class 5 pathogenic. Given our RNA-based findings, carriers of this *MSH6* variant were advised in post-test genetic counseling that this variant is likely pathogenic and that they may receive revised genetic test reports notifying them of a change in classification in due course. Carriers of this *MSH6* variant alone have opted to continue surveillance screening according to recommendations for *MSH6*-LS.

Prior reports of the *MSH2* c.2006G>T variant were in LS cases from European countries including France and Italy. All living members (generations III and IV) of Family C reside in the USA or Australia, although the matriarch of the 13 branches from whom the *MSH2* c.2006G>T variant likely originated was of German heritage. It is possible that these families share ancestral heritage, which if correct, would suggest the *MSH2* c.2006G>T variant may be more widely disseminated among Europeans. More extensive genotyping across the chromosome 2p21 region in carriers of this variant from these ostensibly unrelated families would be required to demonstrate this. Data from Chromium 10× lr-WGS for a dual-carrier member of this family are made available herein for comparison.

While carriers of a single pathogenic MMR variant have the usual 50% risk of transmission to offspring, the carriers of both *MSH2* and *MSH6* PVs in whom both chromosome 2 alleles are affected have a ~100% risk of transmitting either one of the variants to their offspring. Nevertheless, we found that one son (IV.11) of a carrier of both MMR PVs had not inherited either one as predicted. It was important to confirm the veracity of this finding for accurate genetic diagnostic purposes. Repeated genetic testing on two fresh samples at two independent CLIA-approved genetic testing laboratories confirmed IV.11 was negative for both variants and that his MMR genes were intact, hence he was negative for LS. To determine the mechanistic basis for this instance of unexpected non-transmission, we used a combination of Chromium 10× lr-WGS and targeted linked-read sequencing with haplotype reconstruction. The son inherited one maternal chromosome 2 allele and one inferred paternal allele, ruling out pUPD. Using genotypes uniquely shared with his mother, we reconstructed the maternally inherited Chr2p21 allele in the son. *MSH6* is centromeric to *MSH2* on Chr2p21. We showed that in the centromeric region encompassing *MSH6*, the son had inherited the maternal haplotype that was linked to the *MSH2* c.2006G>T variant in the mother. However, in the more telomeric region around *MSH2*, the son had inherited the maternal allele that was linked to the *MSH6* variant in the mother. The haplotype reconstruction showed a recombination event had occurred proximal to the *MSH2* c.2006G>T location (near or within the intervening *KCNK12* gene) and distal to *MSH6* to form a derived wild-type allele. Thus, we were able to attribute the son’s escape from LS to a recombination event that had occurred between the locations of the *MSH2* c.2006G>T and *MSH6* c.3936_4001+8dup (Intronic) variants between the two maternal alleles.

Finally, cancer gene panel testing revealed additional pathogenic variants in two other (non-LS) genes, *MC1R* and *NBN*, that are independently associated with elevated risks of cancer in the branches of Family C that did not carry either MMR variant. Thus, the members of these branches received negative test results for the familial LS-causing pathogenic variants but some members nevertheless had elevated cancer risks due to the finding of one or both *MC1R* and *NBN* variants, for which they were appropriately counseled. A variant within *PTPRJ* was also identified by exome sequencing and targeted testing in a mother with bilateral ovarian cancer and daughter with NHL. However, this variant was not reported to either family member and its pathogenicity remains uncertain. The finding of several distinct pathogenic variants in one family illustrates the need for caution when limiting genetic screening in cancer-affected families to specific genes or to site-specific variants.

## 5. Conclusions

Detailed study of a large 13-branched cancer-affected family whose history spanned four generations revealed genetic variants in several cancer-associated genes. Several branches of the family were diagnosed with Lynch syndrome due to carriage of the *MSH2* c.2006G>T variant, which was reclassified by others as pathogenic during the course of the 12-year follow-up in this family. Reclassification enabled family members to receive an accurate genetic diagnosis. In one branch carrying the *MSH2* c.2006G>T pathogenic variant, the novel intronic variant *MSH6* c.3936_4001+8dup was introduced by marriage in generation II, such that offspring in generation III carried one or both variants. Functional analyses of RNA in family members confirmed the variable splicing patterns of the *MSH2* c.2006G>T variant and also showed the *MSH6* c.3936_4001+8dup variant caused two apparently fully penetrant, erroneously spliced transcripts. While clinical data were equivocal with respect to the pathogenicity of *MSH6*, in part due to the lack of availability of historic tumor specimens, the functional data provide evidence for the *MSH6* c.3936_4001+8dup variant being pathogenic. We therefore proffer the *MSH6* c.3936_4001+8dup variant should be reclassified as likely pathogenic.

Given the finding of P/LP variants in two distinct LS-associated genes with distinct risks for cancer and risks for LS transmission depended on carriage of one or both, genetic counseling in the family, even among members of the same branch, was individually tailored. The finding of additional genetic variants in other cancer-relevant genes in non-LS branches with various other cancer diagnoses suggests genetic testing in relatives should not be limited to targeted genetic testing of specific pathogenic variants. Consideration should be given to offering a multi-cancer gene panel test to all at-risk relatives in the they carry additional cancer-relevant variants.

## Figures and Tables

**Figure 1 cancers-15-00228-f001:**
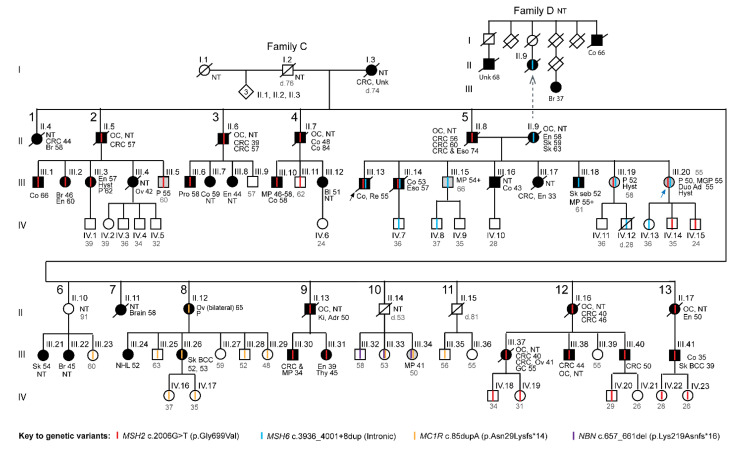
Pedigree showing cancer family history and carriage of cancer-associated genetic variants. Shown are four generations (labeled I–IV) of “Family C” (main figure) and three generation (I–III) of “Family D” (top right), linked by marriage. Family C is a large family carrying the *MSH2* c.2006G>T variant in which additional cancer-associated variants were found. Persons are labeled with the generation and a designated number for reference. Each branch descending from the cancer-affected female in generation I (the likely ancestral carrier of *MSH2* c.2006G>T) is numbered 1–13. Due to the size of the main pedigree, the figure is split with branches 1–5 shown above, and branches 6–13 shown below. All persons in generations I-III are included, however, those who did not consent to join the study are shown as diamonds to preserve privacy. Only members of generation IV who consented to join the study are included in the pedigree, to preserve privacy. Age in years at close of study is given in gray beneath each symbol for those unaffected by cancer or under ongoing follow-up. Age in years at death (d.), where relevant, is provided in gray. All members of generation IV were under 40 years of age at close of study. Closed black symbols represent cancer-affected members with cancer diagnosis given alongside or below as follows: CRC, colorectal cancer; Co, colon; Re, rectal; En, endometrial; Br, breast; Eso, esophageal; Sk, skin (of unknown histology); Seb, sebaceous carcinoma of the skin; BCC, basal cell carcinoma; Ov, ovarian; Ki, kidney, Adr, adrenal gland; GC, gastric; Pro, prostate; Bl, bladder; NHL, non-Hodgkin’s lymphoma; Thy, thyroid. Gray symbols represent a precancerous lesion identified during endoscopic surveillance, as follows: P, colorectal polyp(s); MP, multiple colorectal polyps; MGP, multiple gastric polyps; Duo Ad, duodenal advanced adenoma. Age at diagnosis (number) in years provided in black text alongside or beneath. + indicates multiple diagnoses of polyps above the age shown for first diagnosis. Unk = age at cancer diagnosis unknown. Open (white) symbols represent unaffected members; Colored vertical lines within shapes indicate carriage of a cancer-associated genetic variant, as indicated in the key. OC = obligate carrier of indicated genetic variant, NT = not tested. Hyst indicates females who had undergone a complete hysterectomy and oophorectomy.

**Figure 2 cancers-15-00228-f002:**
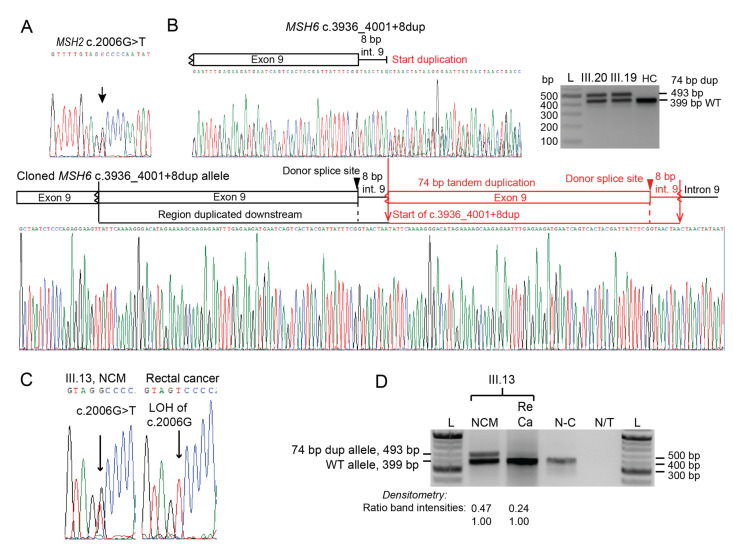
Carriage of both *MSH2* and *MSH6* variants by the proposita and proband and loss of heterozygosity of the wild-type *MSH2* allele in the proband’s rectal tumor. (**A**) Sequence electropherogram across the *MSH2* c.2006G>T variant site in genomic DNA from the proposita, III.20, shows she is a heterozygous carrier. (**B**) Left, sequence electropherograms across the *MSH6* c.3936_4001+8dup (Intronic) variant in the proposita showing a heterozygous frameshift mutation, and beneath, the cloned duplicated variant allele showing the complex structure of the *MSH6* c.3936_4001+8dup (Intronic) variant. Right, photograph of electrophoresis gels showing the size-fractionated PCR fragments across the *MSH6* c.3936_4001+8dup (Intronic) variant in saliva DNA from the proposita (III.20) and her older sister (III.19) showing both are heterozygous for this germline variant (upper fragment). HC, healthy control saliva DNA. (**C**) Sequence electropherogram across the *MSH2* c.2006G>T variant in DNA isolated from formalin-fixed paraffin-embedded normal colorectal mucosa (NCM) and rectal cancer tissues from the deceased proband, III.13, shows complete loss-of-heterozygosity (LOH) of the wild-type allele in the tumor. (**D**) Photograph of electrophoresis gels showing the size-fractionated PCR fragments across the *MSH6* c.3936_4001+8dup (Intronic) variant in genomic DNA from NCM, the rectal cancer (Re Ca) from the proband and peripheral blood lymphocytes from a non-carrier (N-C) of the *MSH6* variant. The fragment intensities from densitometry of the WT and duplicated fragments are shown beneath. The NCM showed reduced amount of the duplicated allele compared to the WT allele, with further reduction of the duplicated allele in the tumor. Original blots see Appendix A.

**Figure 3 cancers-15-00228-f003:**
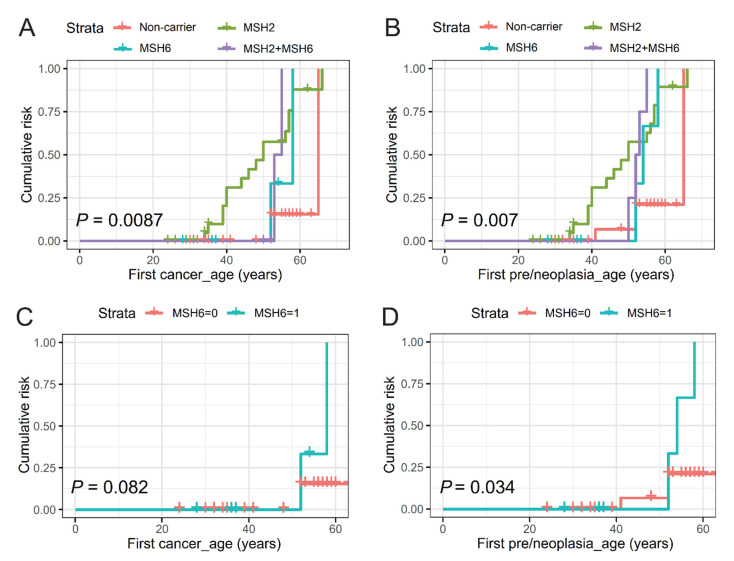
Risk for onset of cancer or pre/neoplasia among confirmed or obligate carriers of *MSH2* c.2006G>T, *MSH6* c.3936_4001+8dup (intronic), or both variants, and non-carriers of either variant. Kaplan–Meier curves are shown of risk by age in years for (**A**,**C**) onset of first cancer of any type, and (**B**,**D**) for onset of first cancer of any type and first colorectal polyp(s) pooled (pre-/neoplasia). For (**A**,**B**), colored lines represent carrier status for the mismatch repair variants; red, carriers of neither variant; green, *MSH2* c.2006G>T alone; teal, *MSH6* c.3936_4001+8dup (Intronic) alone; purple, carriers of both variants. For (**C**,**D**), MSH6 = 0 (red) shows non-carriers of either *MSH2* or *MSH6* variant (*n* = 27), and MSH6 = 1 (teal) shows carriers of *MSH6* c.3936_4001+8dup (Intronic) alone (*n* = 7). *p*-values between groups were obtained by log-rank test.

**Figure 4 cancers-15-00228-f004:**
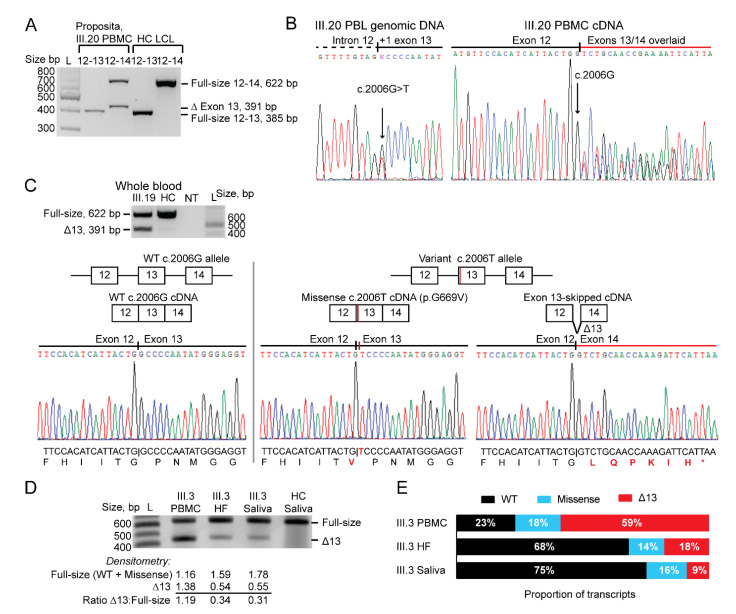
*MSH2* c.2006G>T is associated with both exon 13-skipped and full-sized missense G>T transcripts at different proportions within different tissue types. (**A**) Electrophoretic separation of reverse-transcriptase (RT) PCR products of mRNAs from peripheral blood mononuclear cells (PBMC) from the proposita (carrier of the *MSH2* c.2006G>T variant) and cultured lymphoblastoid cells (LCL) from a healthy control (confirmed non-carrier). RT-PCRs were conducted using the same forward primers within *MSH2* exon 12 and a reverse primer within either exon 13 (385 bp amplicon) or exon 14 (622 bp full-sized amplicon; 391 bp exon 13-skipped amplicon). (**B**) Sequence electropherograms across the c.2006G>T site (black arrows) within the proposita’s genomic DNA (left) and PBMC cDNAs amplified between exons 12–14 (right). The cDNA electropherogram becomes asynchronous from c.2007 due to exon-13 skipping. Only the wild-type G allele was detected at c.2006; the missense T allele was not observed above background noise levels. (**C**) Electrophoretic separation of whole blood *MSH2* exon 12–14 RT-PCR products from the proposita’s sister, III.19 (c.2006G>T carrier) and the same healthy control as in A. Beneath, sequence electropherograms of individual transcripts separated by cloning RT-PCR product, showing the wild-type transcript from the c.2006G allele (left), and mutant transcripts from the c.2006T allele, including a full-length missense transcript (middle) and exon 13-skipped transcript (right). The predicted amino acid sequences for each transcript are shown below the respective electropherogram, with altered amino acid sequences shown in red font. The missense c.2006G>T transcript results in a glycine to valine substitution (p.Gly699Val). The frameshift generated by the exon 13-skipped transcript results in the introduction of a novel sequence of six amino acids followed by a premature stop codon (*). (**D**) Gel electrophoresis and densitometry to quantify relative levels of exon 13-skipped transcripts from full-length (wild-type and missense) transcripts in different tissues from the same carrier, III.3. (**E**) Histogram showing the proportions of the exon 13-skipped, missense c.2006T, and wild-type transcript identified among the cloned RT-PCR (cDNA) products, confirmingvariable levels of the exon 13—skipped versus missense c.2006T transcripts across different tissues from the same carrier, III.3, using another method. These were not identified in the saliva of the same healthy control as in (**A**,**C**). PBMC, peripheral blood mononuclear cells; HF, hair follicles. Original blots see Appendix A.

**Figure 5 cancers-15-00228-f005:**
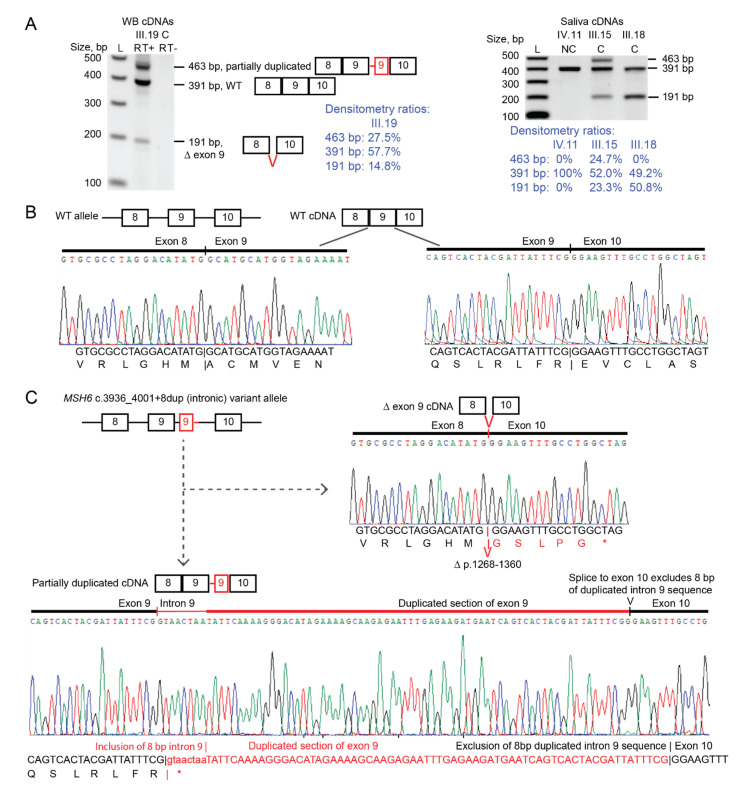
*MSH6* c.3936_4001+8dup (intronic) is associated with two distinct out-of-frame splice variant transcripts. (**A**) Reverse-transcription (RT+)PCR products of *MSH6* exons 8–10 from the saliva of a non-carrier (NC), and the white blood cells (WBC) and saliva (Sa) of *MSH6* c.3936_4001+8dup (intronic) carrier, III.19, after electrophoretic separation show the normal-sized 391 bp fragment derived from the wild-type allele (containing exon 8, 9, and 10) and two distinct aberrant transcripts in WBC: the shorter 191 bp fragment contains exon 8 spliced directly to exon 10, indicating skipping of exon 9; the longer 463 bp fragment contains exons 8, 9, the first 8 nucleotides of intron 9, then the first 64 bp of the duplicated exon 9 sequence up to the point of the duplicated donor splice site, spliced onto exon 10. The final 8 bp of the duplicated intron 9 sequence contained within the genomic *MSH6* c.3936_4001+8dup (intronic) sequence were absent (spliced out) from the longer transcript. Only the exon 9-skipped aberrant transcript was detected in the III.19 saliva. L, 100 bp ladder. RT-, omission of RT enzyme. (**B**) Electropherograms of cloned wildtype *MSH6* transcripts across the expected splice sites derived from the normal allele. (**C**) Partial electropherograms and sequence texts of cloned cDNAs from the two predominant forms of aberrant mRNA transcripts derived from the *MSH6* c.3936_4001+8dup (intronic) allele are shown with the translated amino acid sequences predicted by the ExPASy online Translate tool below, all in IUPAC (International Union of Pure and Applied Chemistry) code. * indicates nonsense codon. The end of the normal exonic sequence text and encoded amino acid reads are indicated by |, after which point the aberrant portions of transcripts begin. Sequence text contained within the c.3936_4001+8dup region are shown in red. Original blots see Appendix A.

**Figure 6 cancers-15-00228-f006:**
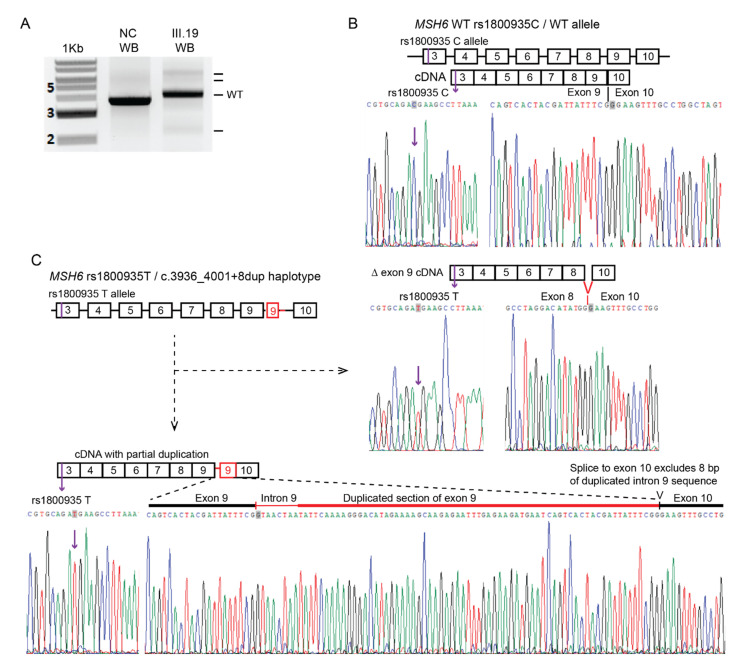
*MSH6* c.3936_4001+8dup (intronic) variant is fully penetrant. (**A**) Electrophoresis gel showing *MSH6* exon 3–10 cDNA fragments from whole blood (WB) RNA samples of a non-carrier control and III.19, dual carrier of the *MSH2* and *MSH6* variants, alongside the 1 Kb Ladder (New England Biolabs, Ipswich, MA, USA). Fragment sizes (kb) for the ladder are shown on the left. As indicated by black lines, the full-length wild-type (WT) fragment (predicted length 3.671 kb) was detected in both individuals, with additional shorter and longer cDNA fragments consistent with altered splicing observed in III.19, as marked by black lines on the right. (**B**,**C**) Illustrative examples of (partial) electropherograms from cloned individual transcripts of III.19 WB cDNAs shown in A, containing the expressed allele of the rs1800935C>T p.D180D (silent) SNP within exon 3, for which III.19 was heterozygous, and which was used to trace the allelic origin of expression of the normal transcripts and the erroneously spliced transcripts. (**B**) The C allele of rs1800935 was linked exclusively to normal transcripts with normal splicing from exon 9 to exon 10. (**C**) The T allele of rs1800935 was linked to the *MSH6* c.3936_4001+8dup variant in genomic DNA. In cDNA, all erroneously spliced transcripts, including the exon 9-skipped transcript (right) and the partially duplicated transcript (below), were both linked to the rs1800935 T allele. No normal transcripts were observed with the rs1800935 T allele, indicating the c.3936_4001+8dup variant allele of MSH6 does not produce normal transcripts.

**Figure 7 cancers-15-00228-f007:**
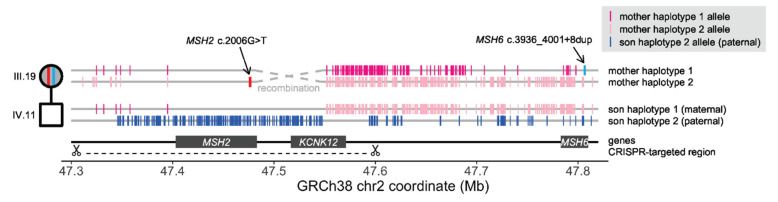
Haplotype reconstruction across Chr2p21-p16 in a mother who carried both *trans* variants *MSH2* c.2006G>T and *MSH6* c.3936_4001+8dup (intronic) shows a recombination event between the two maternal alleles had occurred between the locations of the variants. Haplotypes reconstructed from linked-read whole genome sequence and targeted linked-read sequence data are represented by different shades of pink for maternal haplotypes in female family member, III.19, who is a carrier of both *MSH2* c.2006G>T (red line) and *MSH6* c.3936_4001+8dup (intronic) (teal line) and blue for paternal haplotypes in her eldest son, IV.11, who did not inherit either variant. The locations of the informative polymorphisms used to reconstruct haplotypes are indicated by the pink (maternal genotypes) and blue (paternal genotypes) lines. The location of the recombination between the two maternal haplotypes is indicated by a dashed gray line.

## Data Availability

For reasons of privacy, exome sequence data and linked-read whole genome sequence data has not been made publicly available. These data will be shared upon special request and IRB approval.

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
