# Peer review of "Large Cancer Pedigree Involving Multiple Cancer Genes including Likely Digenic MSH2 and MSH6 Lynch Syndrome (LS) and an Instance of Recombinational Rescue from LS"

_cancers, 2022, doi:10.3390/cancers15010228_

Round 1

Reviewer 1 Report

This is a comprehensive investigation of a large pedigree segregating variants in MSH2, MSH6, and non-Lynch associated genes. RNA studies demonstrated that MSH2 c.2006G>T and MSH6 c.3936_4001+dup both resulted in aberrant splicing, supporting the classification of the former change as a class 5 pathogenic variant and allowing reclassification of the latter change from a VUS to class 4 likely pathogenic variant. Independent segregation of the variants suggested a digenic Lynch syndrome, of which only a few previous reports exist. Interestingly, carriers of both the MSH2 and MSH6 variant did not seem to have any additive risk exceeding that of the MSH2 variant alone. Finally, the demonstration of a genetic recombination-based mechanism of rescue from Lynch syndrome in an offspring of a double variant carrier is novel in the context of Lynch syndrome.   

The study design is informative, and a thorough and careful documentation of the results supports the conclusions drawn.

I only have one main comment: Considering the likely pathogenic interpretation of the MSH6 variant, it is interesting that immunohistochemical analysis of all three polyps tested from two carriers of the MSH6 variant alone showed intact MSH6 expression. While an early or sporadic occurrence of the polyps may serve as an explanation, it is noteworthy that a rectal tumor from a MSH2 and MSH6 variant carrier (III.13) retained the wild-type MSH6 allele whereas the wild-type MSH2 allele was lost (section 3.2.), again arguing against two-hit inactivation in the MSH6-associated tumorigenesis. Would it be possible to study if any of the three polyps had unstable microsatellites as evidence of deficient DNA mismatch repair?

A few minor inconsistencies require revision:

Line 456: “…the proposita’s mother (II.6, Figure 1)”. The pedigree indicates that the proposita’s mother is II.9.

Line 648: “…was predominant over the exon 8-skipped transcript at a ratio of 7:1.” It should apparently be exon 9-skipped transcript.

Line 769: “…consistent with inheritance from his mother (Figure 6), thereby ruling out pUPD in the Chr2p21-p16 region since.” I believe Figure 7 and not Figure 6 illustrates this point.

Reviewer 2 Report

This study used results of a four-generation family and found that not only the specific mismatch repair gene mutations, but also multi-cancer gene panel testing may be necessary to achieve individualized genetic diagnosis and counseling for patients possibly having Lynch syndrome. Furthermore, this study found that specific aberrant products of RNA splicing were associated with specific mutations, but evidence of loss-of-function is unclear. Overall, this study is interesting. However, some concerns should be carefully addressed, as detailed below.

1.     In method section, next generation sequencing procedures and details for a panel of 134 cancer-related genes need to be described. 

2.     How many samples of saliva, peripheral blood, and hair follicles obtained in this study? Please described.

3.     In Figure 2B upper right panel, what does the “HC” mean?

4.     In Figure 4D, using PBMC, hair follicles, and saliva, the authors show that an exon-13 skipping RNA variant of MSH2 was present in a c.2006G>T carrier (#3). This is interesting, however, only saliva from a non-carrier control was used as a reference. To circumvent that RNA variants with exon-13 skipping are not a tissue-specific phenomenon, PBMC, hair follicles, and saliva samples from the healthy non-carrier should be tested. Also, all samples of c.2006G>T carriers available in this study should be tested in parallel.

5.     In Figure 4E, it is difficult to catch the concept the authors wanted to express. Tissue specificity?  

6.     Whether there is any evidence or clues may further provide whether the exon-13 skipping RNA variant of MSH2 affects MSH2 expression or loss of function. In other words, does a heterozygous MSH2 c.2006G allele have a dominant effect?

7.     In Figure 5A left panel, there are no non-carrier controls. Although the label indicates 463 bp, the size of this band is clearly over 300 and less than 400 bp. How can this be explained?

8.     In Figure 5A right panel, #15 and #18 have the same duplication mutation in MSH6, however, why did #18 have no signal at 463 bp? 

9.     Similar to question 6, is there any further evidence or hint as to whether the RNA variants with MSH6 exon-9 partial duplication (the longer transcript) affect MSH6 expression or loss of function? Also, does exon-9-skipped RNA variants of MSH6 (the shorter transcript) affect expression or loss of function of MSH6?

10.  Interpretation of the results presented in Figure 4 and 5 should be careful as there is no evidence that the identified mutations are the cause of the aberrant transcripts. It is recommended to use terms such as “associate” or “correlate”, instead of “result in”.

11.  Please avoid any typos. Some places are missing commas, and some are missing spaces.

Round 2

Reviewer 2 Report

The authors completely answered my concerns. It is now acceptable.